# A Study on the Reliability of Mass, Density, and Fire Performance of Recycled Wastepaper Building Finishing Material Made with Large Wet Cellulose 3D Printers

**Chansol Ahn [1], Dongin Park [2], Jeo Hwang [2] and Dongho Rie [3],***

1   Department of Fire Safety Research, Korea Institute of Civil Engineering and Building Technology, Goyang 18544, Korea
2   Graduate School of Safety Engineering, Incheon National University, Incheon 22012, Korea
3   Fire Disaster Prevention Research Center, Incheon National University, Incheon 22012, Korea
*   Correspondence: riedh@inu.ac.kr

**Abstract:** The impact of non-face-to-face contact following the COVID-19 pandemic has emerged as a social problem and has increased the amount of wastepaper, mainly in home delivery boxes. The appropriate recycling of paper waste is an area where sustainable growth is required in terms of the net environment system and carbon neutrality practice. Therefore, in this study, a specimen of building finishing material using wastepaper was produced using a custom-made large wet cellulose (LWC) 3D printer, and the site applicability of the fire performance was evaluated. The specimen of the building finish material was a mixture of wastepaper and ceramic binder, and the molding of the specimen was uniformly produced by a cylinder injection-type LWC 3D printer. The production reliability of the 3D printer was analyzed by measuring the mass and density of the specimen. The uniformity of the mass and density of the manufactured building finishes were confirmed to have standard deviations of $\pm 0.05$ g and $\pm 0.01$ g/cm$^3$, respectively. The uniformity of the fire performance of specimens was confirmed by checking the relative standard deviation (RSD) value of $\pm 3\%$ under the same ceramic addition conditions from ISO 5660-1. Through the mass and density analysis and fire performance analysis of the building finishing materials, it was confirmed that the same mass, density, and fire performance can be produced simultaneously, and manufacturing using LWC 3D printers has been confirmed to be effective in developing uniform semi-non-combustible and retardant building materials.

**Keywords:** cellulose; 3D printer; flame retardancy; ISO 5660-1; building finishing material

## 1. Introduction

Global home delivery service usage has exploded due to non-face-to-face contact caused by the COVID-19 pandemic in 2020. Changes in the use of home delivery services have led to an increase in paper waste, mainly in home delivery boxes [1]. Korea's home delivery service usage increased by 20.93% in 2020, as compared to 2019 before the COVID-19 outbreak [2]. The average rate of paper waste emissions increased by 15.4% [3], with an increase of 37% in the United States, 32% in the United Kingdom, 11% in Germany, and 20.7% in Japan [4,5]. Table 1 shows the rate of increase in the use of courier services in each country.

**Table 1.** The increase in courier service usage in 2020, compared to 2019.

| Country | Korea | UK | Germany | United States | Japan |
|---|---|---|---|---|---|
| Increase rate | 20.93% | 32% | 11% | 37% | 20.7% |

Sustainable development through recycling is being promoted as a project in each country as an important issue, in the direction of implementing a carbon-neutral climate change response. A project led by the United States National Industrial Competency of Energy, Environment, and Economy (NICE3) to reduce paper waste aimed to create 1.4 million jobs and raise USD 200 billion a year [6]. In 2020, the American Forest and Paper Association (AF&PA) managed to recycle 65.7% of paper consumed in the United States in 2020 [7]. To cope with climate change, Europe promoted the COST ACTIONE 48 project that established a market for the recycling of printing and copying paper through the supply and distribution system and the quality development of recycled paper [8]. In a study on paper recycling, Suseno, N. mixed wastepaper with rice straw, and proceeded with the process of soda pulping to produce recycled paper with a tensile strength of 8.65 N/mm$^2$ under 1:1 mixing conditions [9]. Raut et al. produced an energy-absorbing lightweight brick made by mixing recycled paper mill and cement [10]. Pathak recycled paper mill to manufacture insulation, and confirmed the thermal conductivity of $(0.4 \pm 0.2)$ W/m·K [11]. Astraskas conducted research on paper recycling in various ways, including mixing paper sludge with clay to produce sound absorption composite panels and securing a sound absorption coefficient of 0.59 through ISO 10534-2 [12]. However, the use of paper waste as an indoor building finishing material to increase the added value of paper waste is required, as is the development of a flame-retardant or semi-non-combustible building finishing material to minimize damage to life and property from indoor fires. Currently, building finishing materials commonly used in domestic and foreign buildings are combustible materials, such as flammable aluminum cladding panels, as well as EPS and Urethane foam, which are the main causes of flame spread. Table 2 shows the related building fire accidents.

**Table 2.** Fire accidents caused by finishing materials.

| Date | Casualties | | Fire Location | Country | Reference No. |
|------|-------|--------|---------------|---------|------------------|
|      | **Death** | **Injury** | | | |
| 2015. 12 | - | 15 | The address downtown | Dubai (UAE) | [13] |
| 2016. 06 | 79 | - | Ramat Gan (a residential high-rise building) | Israel | [14] |
| 2017. 06 | 80 | 70 | Granfell Tower | UK | [15] |
| 2017. 12 | 29 | 37 | Jecheon Sports Center | Korea | [16,17] |
| 2019. 02 | - | 1 | Neo 200 building | Australia | [18] |
| 2020. 04 | 38 | 10 | Icheon warehouse | Korea | [19] |
| 2020. 10 | - | 91 | Ulsan mixed-use apartment | Korea | [20] |

To ensure the safety of buildings, countries are making efforts to ensure flame-retardant and semi-non-combustible performance when selecting indoor finishing materials, with the aim of minimizing the spread of fire by finishing materials. Underwriter's Laboratories of Canada (ULC)'s Fire Commission standardized the ISO 5660-1 experiment as a standard test method for evaluating the combustibility of building materials in 1992 [21]. Europe introduced the EN 13823 test method and established a flame-retardant classification based on EN 13501-1 [22]. In Korea, the indoor finishing materials of buildings were classified into non-flammable materials, semi-non-combustible, and flame-retardant materials. The experimental methods for fire performance standards are specified as ISO 5660-1 and ISO 2271 tests [23]. To ensure the safety of construction materials, various studies on the development of flame-retardant and semi-non-combustible construction materials have been conducted based on ISO 5660-1 experiments [24–28]. Nazer and Thavarajah reported that the synergistic effect of phosphorus–nitrogen plays a central role in the treatment of cellulose building materials, such as bamboo and wood, which can increase sustainability as an eco-friendly flame-retardant material [24]. Xu, Zhang, and Wu revealed that by examining the effect of flame retardancy on plywood using zirconium phosphate [a-ZrP], which is a two-dimensional substance, the peak heat release rate (PHHR) and total heat

release (THR) values of plywood were reduced by 41.8 and 22.9%, respectively [25]. Chun, Kim, and Rie revealed that the THR value decreased from 38.63 to 2.50 MJ/m$^2$ when the weight of expanded graphite increased by 30% from the cone calorimeter test on wood particles, expandable graphite, and general binder composites [26]. Beh, Yew, Yew, and Saw conducted a study on thin film moisture absorption coatings, incorporating egg shells (ESs) as flame-preventing bio-filler, confirming that a uniform carbonization layer is provided by reducing the fire propagation effect and heat release rate (HRR) index values in 3.5 and 2.5 wt% of ES bio-filler addition conditions [27]. Li, Zhang, Zuo, Lu, Yuan, and Wu found that sodium silicate-modified fir tree (SSMCF) decreased PHRR and THR values by 58.2 and 49.4%, respectively, compared to conventional fir trees, from a study which set out to improve the properties and flame retardancy of fir trees [28]. Products using 3D printers are used in various fields from the design process for building construction [29,30]. Hwang, Park, Kim, and Rie produced a uniform size of specimen that uses a wet cellulose (WC) 3D printer, demonstrating that ISO 11925-2 experiments could be conducted [31]. In this study, 3D printers were applied to the production, so that wastepaper construction finishes could secure uniform physical properties and fire performance. The production process proceeded in the following order:

1. The grinding of wastepaper;
2. The mixing of ceramic binder with water;
3. Printing with a 3D printer;
4. Drying.

The reliability analysis of the manufacturing process was confirmed by measuring the mass and density of the specimens, and the fire performance of the building finishing material was confirmed through the ISO 5660-1 heat release rate experiment. Figure 1 shows a flow chart of the specimen production and experimental order.

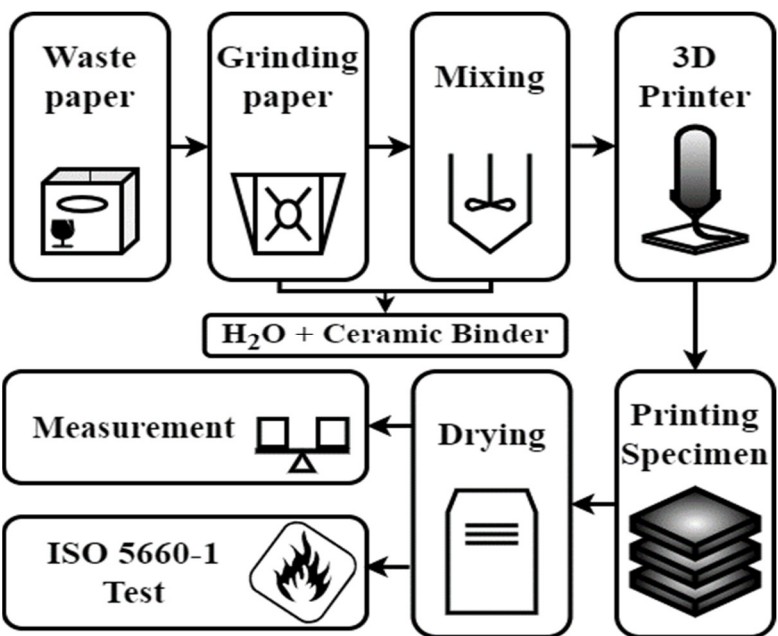

**Figure 1.** Flow chart of the specimen production and experimental order.

## 2. Making a Specimen and Physical Characteristics

### 2.1. Wet Cellulose 3D Printer

For specimen production, the self-produced large wet cellulose (LWC) 3D printer was used to secure specimen uniformity. Table 3 shows the specifications of the 3D printer, and Figure 2 shows a detailed configuration of the 3D printer.

**Table 3.** Specifications of the 3D printer.

| Mechanical | | |
|---|---|---|
| Main body size | 1.4 m × 1.4 m × 0.7 m | |
| Motor frame Size | Axis motor | 57 mm |
| | Extrusion motor | 57 mm |
| Motor step angle | Axis motor | 1.8° (±5%, full step) |
| | Extrusion motor | 1.8° (±5%, 2-Phase) |
| Power specification | 10 A, 250 V | |
| **Printing** | | |
| Zero adjustment | X, Y axes | Limit Sensor |
| | Z axis | BL Touch sensor |
| Nozzle diameter | 10 mm | |
| Extrusion method | Use of cylinder upper lead screw rotational force | |
| **Software** | | |
| Program | Design 123D, Cura 15.04.6 | |
| File format | G-code, STL | |

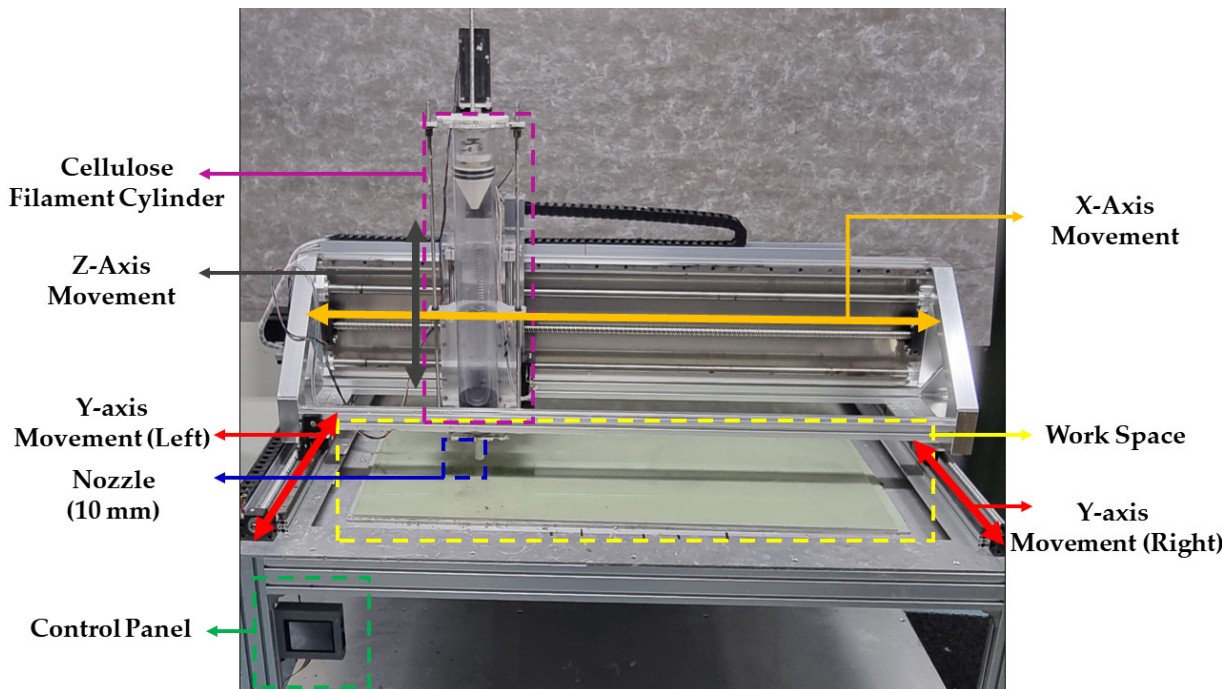

**Figure 2.** Large wet cellulose (LWC) 3D printer and main structure.

*2.2. Cellulose Filament Production*

The cellulose filaments used in printing were ground by mixing wastepaper with water. Figure 3 shows the grinder and the grinding result used for the grinding work. Thereafter, wet-pulverized wastepaper, ceramic binder, and polyacrylamide (PAM) were mixed to manufacture a wet filament for the 3D printer. The chemical compositions of the ceramic binder for flame-retardant performance were 48% MgO, 17% $SiO_2$, 14.6% $Al_2O_3$, 12% Illite, 7% $ZrO_2$, 1% $Al_2O_3$, 0.8% $Fe_2O_3$, and 0.6% CaO. The amount of ceramic binders added to implement flame-retardant performance increased by up to 50% at a rate of 10% by weight of the wastepaper, separated into five types. The amount of PAM added was set

to a constant value of 5% to effectively achieve the uniform extrusion of 3D printing. The water content was increased in proportion to the ceramic binder content to maintain the constant extrusion rate of the filament.

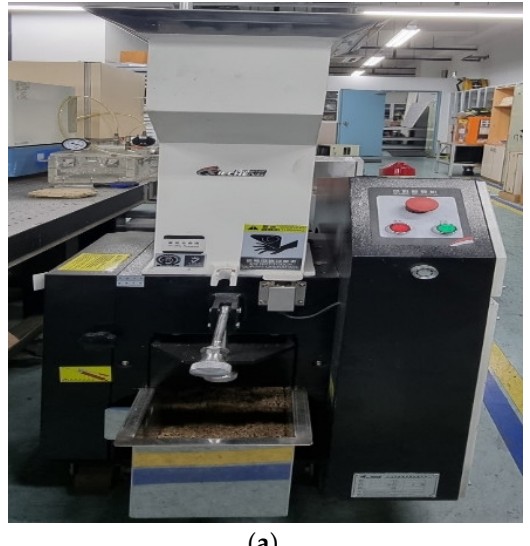
(**a**)

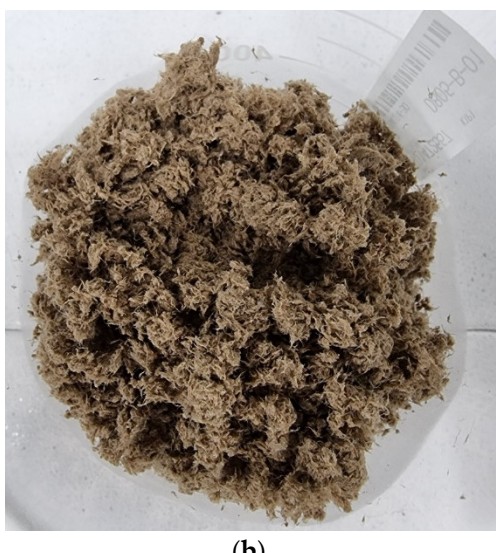
(**b**)

**Figure 3.** (**a**) Grinder equipment. (**b**) Paper grinding result.

Table 4 shows the specimen composition made of flame-retardant building finish.

**Table 4.** The composition of wet cellulose 3D printer filament (g (wt%)).

| Specimen Name | Basic Material | Flame Retardant | Wetting Agent | Flocculent |
|:---:|:---:|:---:|:---:|:---:|
| | Wastepaper | Ceramic Binder | Water | PAM |
| A | 100 (100) | 0 (0) | 500 | |
| B | 100 (83.34) | 20 (16.67) | 750 | |
| C | 100 (76.92) | 30 (23.07) | 800 | 5 |
| D | 100 (71.42) | 40 (28.57) | 850 | |
| E | 100 (66.67) | 50 (33.33) | 900 | |

### 2.3. Wet Cellulose (WC) 3D Printing

The design of the specimen used the 'Design123' software to design the shape of specimen, and to generate the STL file. A generated STL file was converted to a G-code by setting output conditions, such as the filling amount of a cellulose specimen, and the transfer speed of a 3D printer nozzle part using the Cura 15.04.6 program. In the printing process of the specimen, the zero point of the Z-axis of the LWC 3D printer nozzle part was corrected in accordance with the G-code setting value, and the output of the three specimens was simultaneously progressed at a size of 100 mm × 100 mm × 30 mm. The printed specimen was dried for 48 h in a 60 °C dryer. Table 5 shows images of before drying, the drying process, and after drying of the printing specimen.

**Table 5.** Printed and after drying specimen.

| Category | Before Drying | Drying | After Drying |
|---|---|---|---|
| Specimen (ISO 5660-1) | 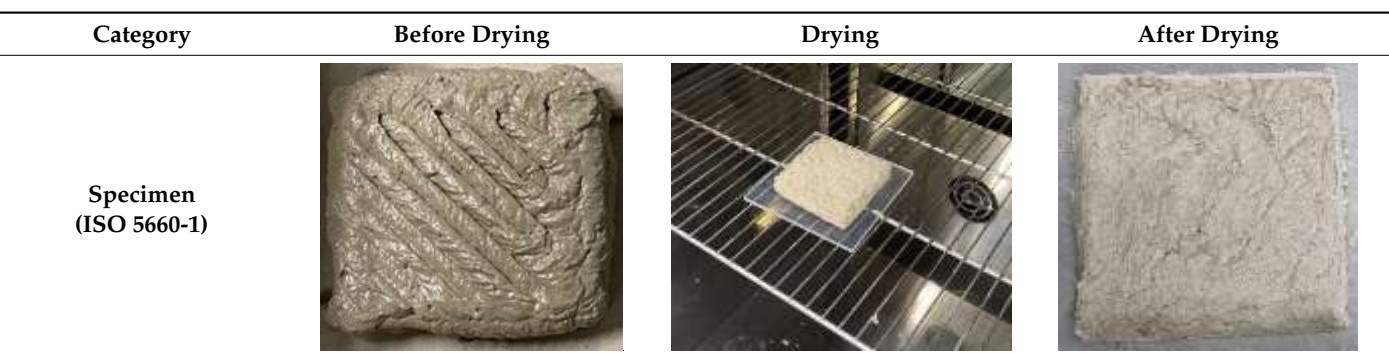 | | |

Table 6 shows the mass and density before and after drying at the time of printing with an ISO 5660-1 test 3D printer, according to a change in the amount of added ceramic binder.

**Table 6.** Physical properties of the ISO 5660-1 specimen.

| Specimen Name | Specimen No. | Weight (g) | | Density (g/cm$^3$) | |
|---|---|---|---|---|---|
| | | Before Drying | After Drying | Before Drying | After Drying |
| A | 1 | 317 | 60 | 0.87 | 0.25 |
| | 2 | 321 | 60 | 0.88 | 0.26 |
| | 3 | 319 | 59 | 0.88 | 0.24 |
| B | 1 | 317 | 60 | 0.77 | 0.21 |
| | 2 | 323 | 60 | 0.72 | 0.22 |
| | 3 | 326 | 60 | 0.82 | 0.23 |
| C | 1 | 380 | 74 | 0.88 | 0.27 |
| | 2 | 370 | 73 | 0.86 | 0.27 |
| | 3 | 382 | 76 | 0.88 | 0.26 |
| D | 1 | 385 | 87 | 0.89 | 0.34 |
| | 2 | 381 | 86 | 0.88 | 0.33 |
| | 3 | 378 | 86 | 0.81 | 0.34 |
| E | 1 | 389 | 88 | 0.86 | 0.32 |
| | 2 | 384 | 87 | 0.85 | 0.31 |
| | 3 | 384 | 89 | 0.87 | 0.33 |

Table 7 shows the standard deviation of the mass and density values of the specimens after 3D printer printing according to the ceramic binder mixing amount.

The standard deviation of the mass of the produced specimen was 1.6 g or less after drying, and the standard deviation of density was found to be less than 0.05 and 0.01 g/cm$^3$ before and after drying, respectively. It was confirmed that the mass and density values of the ISO 5660-1 specimen manufactured by the 3D printer were uniform with respect to the change in the mixing ratio of the mixed materials used in the specimen after drying.

**Table 7.** The ISO 5660-1 specimen standard deviation of mass and density.

| Specimen Name | Standard Deviation | | | |
|---|---|---|---|---|
| | Weight (g) | | Density (g/cm$^3$) | |
| | Before Drying | After Drying | Before Drying | After Drying |
| A | 2 | 0.57 | 0.005 | 0.01 |
| B | 4.58 | 0 | 0.05 | 0.01 |
| C | 6.42 | 1.52 | 0.01 | 0.005 |
| D | 3.51 | 0.57 | 0.04 | 0.005 |
| E | 2.88 | 1 | 0.01 | 0.01 |

### 2.4. Description of the Heat Release Rate Test

Since a performance certification procedure for the heat release rates to build finish materials must be conducted to ensure safety in a fire accident, the ISO 5660-1 cone calorimeter test method was applied in this study to measure the heat release rate of the test specimen. Specifications for the ISO 5660-1 specimens were 100 mm in width, 100 mm in length, and 50 mm or less in thickness, and the specimens used in this test were 100 mm in width, 100 mm in length, and 30 mm in thickness. The test specimen was managed in a temperature-controlled humidifier for 2 days under the conditions of $23 \pm 2\,^\circ$C and $50 \pm 5\%$ relative humidity, as defined in ISO 5660-1. The test was based on the principle that radiation heat is continuously applied to the specimen located at a distance of 22.5 mm from the heater at a heat flow rate of 50 kW/m$^2$. The calculation of the HRR was based on the principle that about 13.1 MJ of heat is generated when 1 kg of oxygen is consumed when a material is burned. The PHRR represents the HRR at the time when the amount of oxygen consumed during combustion is the highest and the THR value is the value expressed by integrating the HRR. The amount of oxygen consumption required to obtain the HRR is measured through the oxygen analyzer of the cone calorimeter [32]. Figure 4 shows the thermostatic humidifier and the ISO 5660-1 cone calorimeter (FESTEC Co., Korea) used in this experiment.

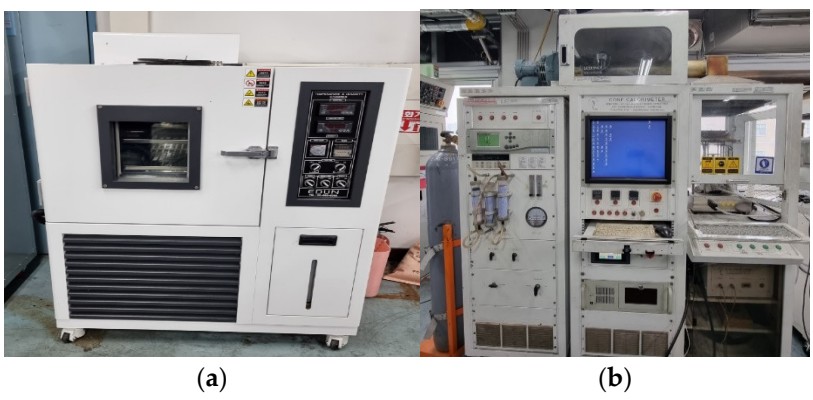

(**a**)       (**b**)

**Figure 4.** (**a**) The thermostatic humidifier. (**b**) The International Organization for Standardization 5660-1 (cone calorimeter).

Table 8 presents the ISO 5660-1 test flame-retardant material and quasi-non-combustible performance standards, as specified by the International Organization for Standardization, and applies the evaluation criteria for the specimen's flame-retardant material and semi-non-combustible experiments [32]. In this study, the ISO 5660-1 test was conducted up to 600 s, which is the semi-non-combustible performance test standard, to confirm whether the specimen meets such standards.

**Table 8.** Performance standard using the ISO 5660-1 test method.

| Standard | Class | Evaluation Criteria |
|---|---|---|
| ISO 5660-1 (cone calorimeter method) | Semi-non-combustible material | The total radiant heat 10 min after heating is 8 MJ/m$^2$. Within 10 min, the max. heat radiant rate does not exceed 200 kW/m$^2$ for longer than 10 consecutive seconds. There shall not be crack that penetrates sample, hole, or melting (for mixed content materials, which includes the melting and dissipation of all core materials) after heating for 10 min. |
| ISO 5660-1 (cone calorimeter method) | Fire-retardant material | Total radiant heat 5 min after heating is 8 MJ/m$^2$. Within 5 min, the max. heat radiant rate does not exceed 200 kW/m$^2$ for longer than 10 consecutive seconds. There shall not be crack that penetrates sample, hole, or melting (for mixed content materials, which includes the melting and dissipating of all core materials) after heating for 10 min. |

## 3. Result of the ISO 5660-1 Test

Figures 5 and 6 show the same additional condition ISO 5660-1 experiments in triplicate of the fire performance for the HRR and THR values of the specimens. Figure 5 shows the HHR of the three specimens for the same additional conditions of the ceramic binder, while Figure 6 shows the time-specific variation of THR values of three specimens at the same ceramic binder addition conditions. Table 9 shows the respective values of PHRR, THR at 300 s, THR at 600 s, and standard deviation for the five ceramic binder addition conditions.

**Table 9.** ISO 5660-1 test results of PHRR, THR at 300 s, THR at 600 s, and standard deviation.

| Name. | No. | PHRR (kW/m$^2$) | Standard Deviation (PHRR) | THR (MJ/m$^2$) at 300 s | THR (MJ/m$^2$) at 600 s | Standard Deviation (THR) at 300 s | Standard Deviation (THR) at 600 s |
|---|---|---|---|---|---|---|---|
| A | 1 | 160.42 | | 27.2 | 55.1 | | |
| | 2 | 145.65 | 11.44 | 25.4 | 53.0 | 1.17 | 1.06 |
| | 3 | 137.9 | | 25.0 | 53.7 | | |
| B | 1 | 79.89 | | 14.0 | 27.3 | | |
| | 2 | 79.08 | 2.64 | 14.3 | 28.5 | 0.25 | 0.79 |
| | 3 | 74.96 | | 13.8 | 27.0 | | |
| C | 1 | 51.05 | | 9.8 | 19.1 | | |
| | 2 | 50.79 | 2.10 | 9.8 | 20.1 | 0.17 | 0.52 |
| | 3 | 54.56 | | 10.1 | 19.9 | | |
| D | 1 | 47.11 | | 9.2 | 17.6 | | |
| | 2 | 43.96 | 1.58 | 9.1 | 17.9 | 0.1 | 0.40 |
| | 3 | 45.19 | | 9.3 | 18.4 | | |
| E | 1 | 23.18 | | 4.8 | 10.2 | | |
| | 2 | 22.57 | 5.09 | 4.8 | 10.1 | 0.34 | 0.05 |
| | 3 | 31.68 | | 5.4 | 10.2 | | |

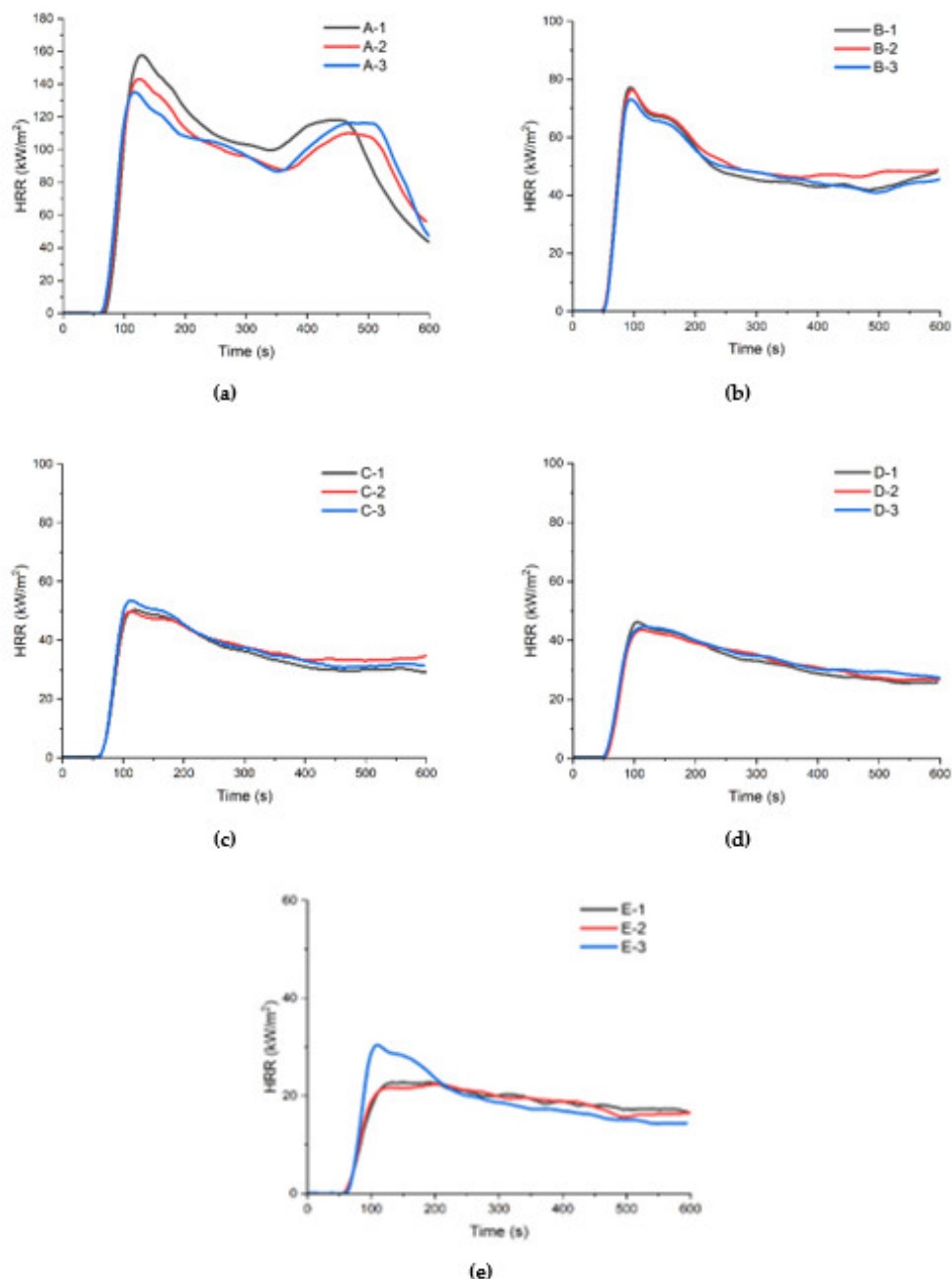

**Figure 5.** HRR measurement of all specimens: (**a**) ceramic—0 g, (**b**) ceramic—20 g, (**c**) ceramic—30 g, (**d**) ceramic—40 g, (**e**) ceramic—50 g.

The PHHR and THR results confirm the improvement in fire performance of the test specimen due to the increase in ceramic binder content. It was confirmed that the performance of flame-retardant material was secured by confirming that the THR value at 300 s was 8 MJ/m$^2$ or less for all three specimens under the condition that 50 g of ceramic binder was added based on 100 g of waste paper.

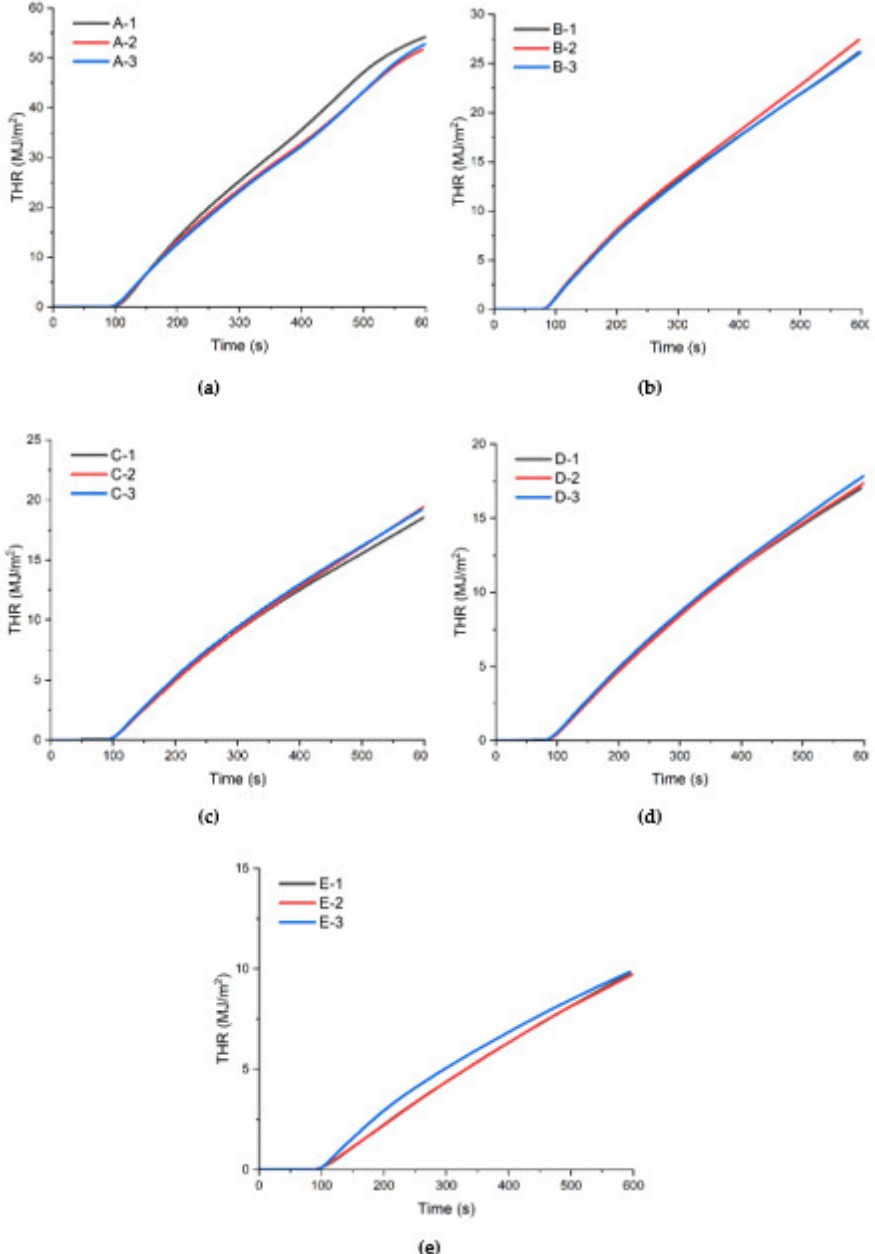

**Figure 6.** THR measurements of all specimens: (**a**) ceramic—0 g, (**b**) ceramic—20 g, (**c**) ceramic—30 g, (**d**) ceramic—40 g, (**e**) ceramic—50 g.

## 4. Discussion

The fire performance reliability of the recycled wastepaper building materials using the LWC 3D printer was verified by calculating the relative standard deviation (RSD) of Equation (1):

$$Relative\ Standard\ Deviation\ (RSD)\ [\%] = \left( \frac{Standard\ Deviation}{Average} \right) \times 100, \qquad (1)$$

Table 10 shows the average, standard deviation (SD), and relative standard deviation (RSD) of PHHR. Figure 7 shows the distribution of the PHHR, the average value, and the tendency of the PHHR due to the change in ceramic binder content.

**Table 10.** ISO 5660-1 result of PHRR average, SD, and RSD.

| Specimen Type | Average (kW/m$^2$) | SD (kW/m$^2$) | RSD (%) |
|---|---|---|---|
| A | 147.99 | 11.44 | 7.73 |
| B | 77.97 | 2.64 | 3.39 |
| C | 52.13 | 2.10 | 4.03 |
| D | 45.42 | 1.58 | 3.49 |
| E | 25.81 | 5.09 | 19.72 |

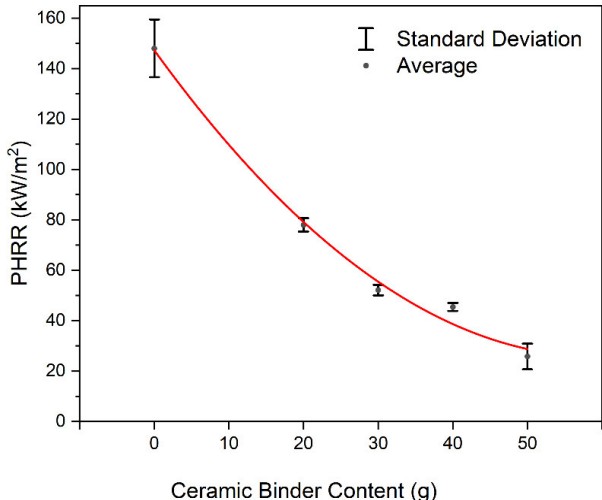

**Figure 7.** PHRR measurement and standard deviation of specimens.

Equation (2) establishes the trend line for PHHR by the amount of added ceramic binder:

$$y = 0.0345x^2 - 4.0977x + 147.3, \tag{2}$$

where $y$: PHRR (kW/m$^2$) and $x$: ceramic binder content (g).

Table 11 shows the average THR, SD, and RSD values in ISO 5660-1 experiment for 300 s, while Figure 8 shows the distribution of THR at 300 s, Average, and the trend line of THR at 300 s with respect to the change in ceramic binder content. The dotted line represents 8 MJ/m$^2$, which is the flame-retardant performance standard THR of the ISO 5660-1 experiment.

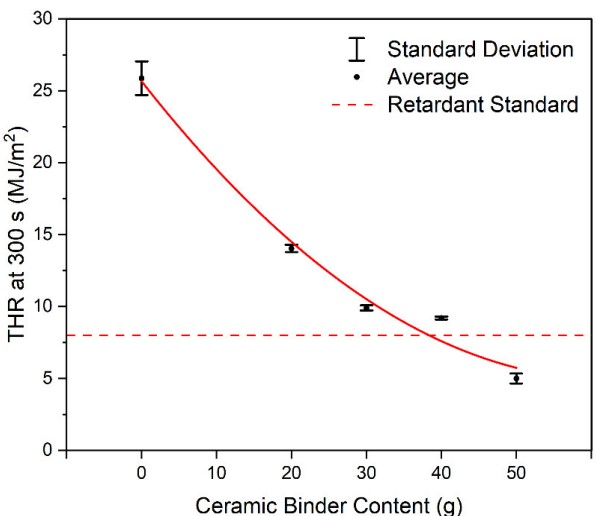

**Figure 8.** THR at 300 s measurement and the standard deviation of specimens.

**Table 11.** ISO 5660-1 results of the average THR (at 300 s), SD, and RSD values.

| Name. | Average (MJ/m$^2$) | SD (MJ/m$^2$) | RSD (%) |
|---|---|---|---|
| A | 25.86 | 1.17 | 4.53 |
| B | 14.03 | 0.25 | 1.79 |
| C | 9.9 | 0.17 | 1.74 |
| D | 9.2 | 0.1 | 1.08 |
| E | 5 | 0.34 | 6.8 |

Equation (3) shows the found trend line to THR at 300 s for the amount of ceramic binder addition change:

$$y = 0.0053x^2 - 0.6649x + 25.66, \tag{3}$$

where $y$: THR at 300 s (MJ/m$^2$) and $x$: ceramic binder content (g).

Table 12 shows the average THR, SD, and RSD values in the ISO 5660-1 experiment after 600 s, while Figure 9 shows the distribution of THR in 600 s, the average, and the trend line due to changes in ceramic binder additions. The dotted line represents 8 MJ/m$^2$, which is the semi-non-combustible performance standard THR of the ISO 5660-1 experiment.

**Table 12.** ISO 5660-1 results of the average THR (at 600 s), SD, and RSD values.

| Name. | Average (MJ/m$^2$) | SD (MJ/m$^2$) | RSD (%) |
|---|---|---|---|
| A | 53.93 | 1.06 | 1.98 |
| B | 27.6 | 0.79 | 2.87 |
| C | 19.7 | 0.52 | 2.68 |
| D | 17.96 | 0.40 | 2.24 |
| E | 10.16 | 0.05 | 0.56 |

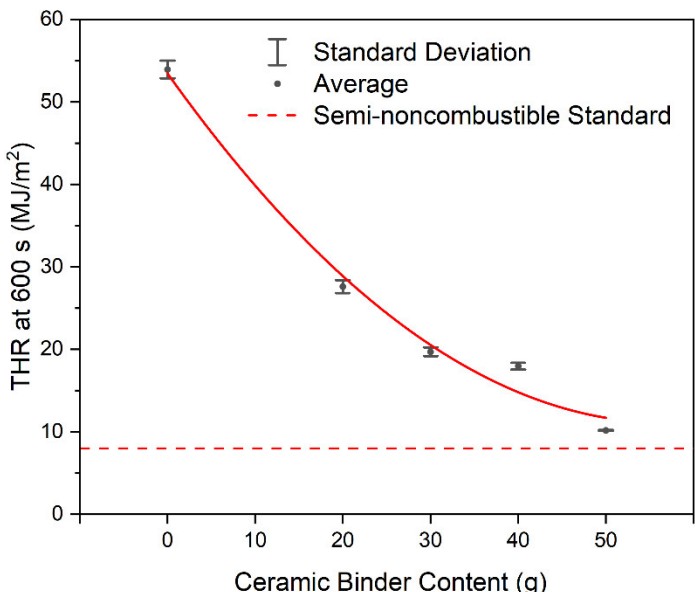

**Figure 9.** THR measurements at 600 s and the standard deviation of specimens.

Equation (4) shows the found trend line to the THR value at 600 s for changes in the amount of added ceramic binder:

$$y = 0.0131x^2 - 1.4912x + 53.462, \tag{4}$$

where $y$: THR at 600 s (MJ/m$^2$) and $x$: ceramic binder content (g).

### 5. Conclusions

The following conclusions were reached through the manufacture of wastepaper–ceramic binder mix materials, the manufacture of specimens using a self-developed wet 3D printer, the mass and density analysis of specimens produced, and the ISO 5660-1 experimental results.

First, the increase in the amount of added ceramic binder resulted in an increase in the density of the product after drying.

Second, a mass standard deviation of $\pm 0.05$ g and a density standard deviation of $\pm 0.01$ g/cm$^3$ after drying the specimens made with a self-developed LWC 3D printer revealed that the specimens can be made with the same physical property value in a uniform amount.

Third, the RSD measurement of fire performance through the ISO 5660-1 test confirmed that all added ceramic binder conditions show RSD values within 3% in THR at 600 s, signifying the end of the test, and revealed that a 3D printer can produce specimens with uniform fire performance.

Fourth, the ISO 5660-1 experimental results confirmed that with increasing added ceramic binder, the PHHR decreases and does not exceed 200 kW/m$^2$ for 10 consecutive seconds in all test specimens. As the content of ceramic binder increases, the THR shows a decrease of about 16%. Specimens with conditions for adding 50 g of wastepaper based on 100 g of ceramic binder were verified to meet the performance standards of fire-retardant materials by checking the measured values below the THR of 8 MJ/m$^2$ based on 300 s.

Fifth, by adding ceramic binder, the THR of the cellulose building finishing material can show a trend equation of $y = 0.0131x^2 - 1.4912x + 53.462$, and a correlation between the mixing conditions of the ceramic binder and the heat release amount was confirmed.

Finally, it was confirmed that specimens produced by 3D printers under the condition of adding 50 g of ceramic binder to 100 g of wastepaper can meet fire safety standards and can therefore be used as building finishing materials. Furthermore, LWC 3D printer manufacturing technology was confirmed to be useful for material development by ensuring uniformity in physical properties.

**Author Contributions:** Conceptualization, D.R.; methodology, D.R.; software, D.P.; validation, D.P. and J.H.; analysis, C.A. and D.P.; resources, D.P.; writing, D.P. and J.H.; review and editing, D.R. and D.P.; visualization, D.P. and J.H.; supervision, D.R.; project administration, D.R.; funding acquisition, C.A. and D.R. All authors have read and agreed to the published version of the manuscript.

**Funding:** This work was supported by an Incheon National University Research grant in 2021.

**Conflicts of Interest:** The authors declare no conflict of interest.

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
