# Peer review of "A Study on the Reliability of Mass, Density, and Fire Performance of Recycled Wastepaper Building Finishing Material Made with Large Wet Cellulose 3D Printers"

_sustainability, doi:10.3390/su142013090_

Round 1

Reviewer 1 Report

This paper evaluates the fire performance of a specimen of building finishing material produced using a custom-made Large Wet Cellulose 3D printer. The production reliability of the 3D printer is analyzed by measuring the mass and density of the specimen. The efficiency of manufacturing using LWC 3D printers is confirmed through the mass and density analysis and fire performance analysis. It is shown that LWC 3D printers are effective in developing semi-non-combustible materials. The results obtained in the paper are reliable and can be used in practise.  I recommend the paper for publication.  

Author Response

Dear Reviewer

Thank you for your kind review.

Thank you.

Reviewer 2 Report

The analysis of the submitted article showed several claims that are not completely clear, so please offer some more detailed explanations. But, considering the small number of samples and tested properties, I do not see a significant scientific contribution.

Author Response

Dear Reviewer Thank you for your careful review, and we will send you the answers to your questions as attachments.

Reviewer 3 Report

The authors have highlighted the recent increase of wastepaper used by home delivery boxes, which can cause a series of environmental and economic issues. Based on systematic investigations, this work develops a novel building finishing material using wastepaper using a custom-made Large Wet Cellulose (LWC) 3D printer, and the site applicability of the fire performance was evaluated. I believe this research has delivered a good idea for developing functional and eco-friendly materials. Therefore, I suggest this manuscript can be accepted after addressing the following questions.

1.       What is the density and detailed chemical composition of ceramic binder content? How the addition of the authors has brought influenced the density of the final product? Please add discussion accordingly.

2.       The PHHR and THR results confirm the improvement in fire performance of the test specimen due to the increase in ceramic binder content. I am wondering if this fire performance is compatible with the mainstream/popular building finishing materials on the market.

Author Response

(The authors gave the same response as above.)

Reviewer 4 Report

1. The last paragraph of the abstract--Through the mass and density analysis and fire performance analysis of the building finishing materials, the efficiency of manufacturing using LWC 3D printers has been confirmed to be effective in developing uniform flame retardant and semi-non-combustible materials. How to find out its affection? Please modify them.

2. Is there any special consideration for the ratio of Wastepaper and Ceramic Binder specimen preparation in Table 4?

3. About the results of ISO 5660-1 Test. The three parameters of HRR, THR and PHRR used in the results should be explained in detail and how to obtain them?

4. The dotted lines in Figure 8 and Figure 9 are Retardant Standard and Semi-non-combustible Standard respectively, but neither seems to be explained in the article? It can make a little difficult for readers to understand.

Author Response

(The authors gave the same response as above.)

Round 2

Reviewer 2 Report

I have no further requests.